# Comparative Study of Different H_2_S Donors as Vasodilators and Attenuators of Superoxide-Induced Endothelial Damage

**DOI:** 10.3390/antiox12020344

**Published:** 2023-02-01

**Authors:** Elisabetta Marini, Barbara Rolando, Federica Sodano, Federica Blua, Giulia Concina, Stefano Guglielmo, Loretta Lazzarato, Konstantin Chegaev

**Affiliations:** 1Department of Drug Science and Technology, University of Turin, 10125 Turin, Italy; 2Department of Pharmacy, “Federico II” University of Naples, 80131 Naples, Italy; 3Rita Levi-Montalcini Department of Neuroscience, University of Turin, 10125 Turin, Italy

**Keywords:** hydrogen sulfide, hydrogen sulfide-donors, vasodilation, pyrogallol, superoxide anion, oxidative stress, vascular endothelium

## Abstract

In the last years, research proofs have confirmed that hydrogen sulfide (H_2_S) plays an important role in various physio-pathological processes, such as oxidation, inflammation, neurophysiology, and cardiovascular protection; in particular, the protective effects of H_2_S in cardiovascular diseases were demonstrated. The interest in H_2_S-donating molecules as tools for biological and pharmacological studies has grown, together with the understanding of H_2_S importance. Here we performed a comparative study of a series of H_2_S donor molecules with different chemical scaffolds and H_2_S release mechanisms. The compounds were tested in human serum for their stability and ability to generate H_2_S. Their vasorelaxant properties were studied on rat aorta strips, and the capacity of the selected compounds to protect NO-dependent endothelium reactivity in an acute oxidative stress model was tested. H_2_S donors showed different H_2_S-releasing kinetic and produced amounts and vasodilating profiles; in particular, compound **6** was able to attenuate the dysfunction of relaxation induced by pyrogallol exposure, showing endothelial protective effects. These results may represent a useful basis for the rational development of promising H_2_S-releasing agents also conjugated with other pharmacophores.

## 1. Introduction

Hydrogen sulfide (H_2_S), together with carbon monoxide (CO) and nitric oxide (NO), belong to the group of gaseous signaling molecules or “gasotransmitters”. These three species play a pivotal role in many pathophysiological processes of the cardiovascular system [1,2,3,4].

Although NO is still considered the major gaseous vasodilator, the identification of the biological importance of H_2_S has aroused increasing interest in its role [3,5,6]. The ongoing research clearly shows that H_2_S is an important independent mediator [7,8,9], as well as an enhancer of NO-mediated effects on the cardiovascular system [10,11]. Over the past decade, it has been progressively demonstrated that H_2_S, either endogenously produced or intentionally administered using H_2_S-donating compounds, is able to influence a wide range of physiological and pathophysiological processes [12]. Indeed, it takes part in the homeostatic regulation of cardiovascular, respiratory, gastroenteric, nervous, immune, and endocrine systems [1,13].

Nowadays, increasing research evidence has corroborated the protective effects of H_2_S in cardiovascular diseases (CVD) [14,15,16,17,18,19,20,21,22] such as cardiac hypertrophy, heart failure, myocardial ischemia/reperfusion (I/R) injury [23], hypertension [16,24] and atherosclerosis [25]. H_2_S reveals its protective potential by acting as an activator of angiogenesis [26], as a basal vasorelaxant agent [27], and as a blood pressure and heart rate regulator [28,29]. It has also been demonstrated that the mechanisms of these cardioprotective effects implicate antioxidative, anti-inflammatory, and pro-angiogenic behavior, in addition to the inhibition of cell apoptosis, and ion channel regulation [30,31]. In particular, it has been reported that H_2_S-mediated endothelial cells’ protection from oxidative stress could be due to direct antioxidant action, as well as maintenance of mitochondrial structure and function [32].

The first pharmacological tools used for the studies of H_2_S effects were inorganic salts, and in particular, sodium sulfide (Na_2_S) and hydrosulfide (NaHS), which were very helpful in clarifying their physio-pathological roles in mammalian organisms [33].

Although these compounds are commercially available and easy to manipulate, they have some important limitations. In particular, these salts hydrolyze upon reaction with water giving rise to an instantaneous supraphysiological level of H_2_S followed by a quick decrease in its concentration [34]. To overcome this problem, several organic structures able to release H_2_S under physiological conditions have been proposed. H_2_S donors differ in the mechanism of activation and H_2_S release kinetic [33].

Here we report a comparative study of a series of H_2_S donor molecules with extensively varied chemical scaffolds and mechanisms of H_2_S release (Figure 1). Their stability in human serum and ability to generate H_2_S, as well as their vasodilator effects, were evaluated. The capacity of the selected compounds to preserve NO-dependent endothelium reactivity in an in vitro model of acute oxidative stress was also reported.

## 2. Materials and Methods

### 2.1. Synthesis

The synthetic procedures and physicochemical characterization of all studied compounds are reported in Appendix A.

### 2.2. Stability and H_2_S Release

#### 2.2.1. Stability of the Compounds at pH 7.4 in Phosphate Buffer (PBS) in the Absence and Presence of L-Cysteine

The compounds were solubilized at a concentration of 100 µM (starting from a 10 mM stock solution in DMSO) in 0.1 M PBS, pH = 7.4, in the absence or presence of 5 mM L-cysteine (50×). The solutions were incubated at 37.0 ± 0.5 °C, and at appropriate time intervals, aliquots were analyzed by RP-HPLC as described below. All experiments were performed at least in triplicate.

#### 2.2.2. Stability of the Compounds in Human Serum

A solution of each compound (10 mM) in DMSO was added to human serum (obtained by filtration and sterilization from AB, Sigma-Aldrich (Merck KGaA, 64,271 Darmstadt, Germany) male human blood) and was preheated to 37 °C to obtain a final concentration of 200 µM. The resulting solution was incubated at 37.0 ± 0.5 °C, and aliquots of 300 µL were taken at appropriate time intervals and added to 300 µL of CH_3_CN acidified with 0.1% formic acid in order to deproteinize the serum. The mixture was centrifuged for 10 min at 2150× *g*, and the clear supernatant was filtered (PTFE filters, 0.45 µm, VWR (VWR International S.r.l., Milano, Italy)) and analyzed by RP-HPLC as described below. All experiments were performed at least in triplicate.

#### 2.2.3. RP-HPLC Analysis of Stability Assays

RP-HPLC analysis was performed with an HP 1100 chromatograph system (Agilent Technologies, Palo Alto, CA, USA) equipped with an injector (Rheodyne, Cotati, CA, USA), a quaternary pump (model G1311A), a membrane degasser (model G1379A), and a diode-array detector (DAD, model G1315B) integrated into the HP1100 system. The data were processed using an HP ChemStation system (Agilent Technologies, Palo Alto, CA, USA). A stationary phase Tracer Excel 120 ODSB (25 × 0.46, 5 μm; Tecnokroma (Teknokroma Analytical S.A., Barcelona, Spain)) was used. The mobile phase consisted of acetonitrile 0.1% HCOOH/water 0.1% HCOOH at a flow rate of 1.0 mL/min, with gradient elution: 35% acetonitrile up to 5 min, 35 to 80% acetonitrile between 5 and 10 min, 80% acetonitrile between 10 and 20 min. The injection volume was 20 μL. The column effluent was monitored for each compound at the wavelength of maximum absorption (referenced against a 700 nm wavelength). Using this RP-HPLC procedure, we separated the compounds from any degradation products and we quantified the compounds during incubation time. The quantification was performed using a calibration curve obtained with standard solutions of compounds chromatographed under the same experimental conditions, in a concentration range of 1–100 μM (r^2^ > 0.99).

#### 2.2.4. Determination of H_2_S Release in Human Serum (Dansilazide Method)

60 μL of dansylazide solution (10 mM in ethanol) and 40 μL of H_2_S donor compound stock solution (10 mM in DMSO) were added to 1900 μL of human serum prewarmed at 37 °C to obtain an initial compound concentration of 200 μM. The solution was incubated at 37 ± 0.5 °C, and at fixed time (1, 4, and 24 h) 200 μL of the reaction mixture was diluted with 200 μL of CH_3_CN to have a final concentration of compound of 100 μM. The mixture was vortexed, centrifuged (10 min at 2150× *g*) and the clear supernatant was filtered (0.45 μm PTFE filters) and analyzed by RP-HPLC. All experiments were performed at least in triplicate. HPLC analyses were performed with a HP 1200 chromatograph system (Agilent Technologies, Palo Alto, CA, USA) equipped with a quaternary pump (model G1311A), a membrane degasser (G1322A), a UV detector, MWD (model G1365D) and a fluorescence detector (model G1321A) integrated into the HP1200 system. Data analysis was performed using an HP ChemStation system (Agilent Technologies). The sample was eluted on a Tracer Excel 120 ODSB C18 (250 × 4.6 mm, 5 μm; Teknokroma); the injection volume was 20 μL. The mobile phase consisted of 0.1% aqueous HCOOH and CH_3_CN 20/80 *v/v*; elution was in isocratic mode at a flow rate of 1.0 mL/min. The signals were obtained in fluorescence using an excitation and emission wavelength of 340 and 535 nm, respectively, and a gain factor = 10. The values obtained from integrating the dansyl amide peak were interpolated into a calibration curve, prepared using NaHS as a standard so that the concentration of dansyl amide in each sample is an index of the amount of H_2_S.

### 2.3. Functional Experiments

Animals were handled humanely in accordance with recognized guidelines on experimentation; the “3 Rs” policy (99/167/EC: Council Decision of 25/1/99) of replacement by alternative methods, reduction of the number of animals, and the refinement of experiments has been fully applied. The protocol has been approved by Ministero della Salute, “Studio preliminare del profilo farmacocinetico e farmacodinamico di composti di nuova sintesi ad attività multifattoriale”. Responsible: Elisabetta Marini. Cod. n. 56105.N.ZMT, approved on 23 June 2018.

#### 2.3.1. Vasodilating Activity

Vasodilating activity was studied on the thoracic aortas from male Wistar rats (180–200 g), following a protocol published elsewhere [35]. All synthesized compounds were dissolved in DMSO. The addition of the vehicle had no perceptible consequence on contraction level. Results were expressed as EC_50_ ± SE (μM), *n* = 3–8.

#### 2.3.2. Vasoprotection in Rat Aorta with Endothelium Impairment Induced by Pyrogallol

The experiments were performed on rat thoracic aortic rings from male Wistar rats (180–200 g), prepared as previously described [36]. The effect of H_2_S donor or catechin, taken as a reference, on acetylcholine-induced relaxation in aortic rings pre-incubated with pyrogallol was studied as published elsewhere [37], with some modifications. Aortic rings were incubated with diethyldithiocarbamate (DETCA) 10 mM for 1 h to irreversibly inactivate endogenous superoxide dismutase (SOD). Then, the aortic rings were washed and randomly divided into four groups: (1) control: endothelium-intact rings incubated in Krebs’ solution with vehicle only (DMSO); (2) H_2_S donor: endothelium-intact rings incubated with H_2_S donor (100 μM) or reference catechin (10 mM) for 30 min; (3) pyrogallol: endothelium-intact rings incubated with 500 μM pyrogallol for 15 min; (4) H_2_S donor plus pyrogallol: endothelium-intact rings incubated with H_2_S donor (100 μM) or reference catechin (10 mM) for 30 min and 500 μM pyrogallol for the last 15 min. After the incubation time, the rings were washed, 1 μM phenylephrine was added to the organ baths, then acetylcholine (10^−9^–l0^−5^ M) was added cumulatively. Tension was measured and calculated as the percentage of contraction in response to phenylephrine (1 μM). Data were expressed as mean ± SEM, *n* = 3–9.

#### 2.3.3. Data Analysis

All data were expressed as mean ± SEM. Statistical comparisons were evaluated by Student’s *t*-test for unpaired data. *p* values < 0.05 were considered significantly different. The Gaussian distribution of data was verified by the D’Agostino–Pearson normality test. Statistical analyses were performed by GraphPad Prism 7.0 (GraphPad Software Inc., San Diego, CA, USA).

## 3. Results

### 3.1. Stability and H_2_S Release

H_2_S donor acids were first evaluated for their chemical stability under physiological-like conditions at pH 7.4 in a phosphate buffer solution. The results are reported in the Appendix A as % residual concentration of compounds after 24 h of incubation. In these experiments, most of the compounds showed good stability, suggesting the absence of spontaneous H_2_S release at physiological pH. Only compound **7** displayed a reduced residual concentration, attributed to the hydrolytic instability of the N-bezoylsulfanil group. The experiment was repeated in the presence of cysteine to evaluate the stability of the compounds toward reactivity with the sulfhydryl group. The stability profiles of the H_2_S donor in PBS, at three different incubation times, in the presence of an excess of cysteine (50×) are reported in Appendix A.

To understand the potential of these H_2_S-donor substructures, it seemed important to evaluate their behavior in human serum, which is a complex sample, rich in metabolic enzymes, proteins and various endogenous thiols (cysteine, glutathione, homocysteine and –SH groups of proteins), which can profoundly influence the reactivity of these molecules. In these experiments, both stability and H_2_S release of all compounds were evaluated (Table 1). Stability data in human serum show that under these conditions, the carboxylic acids are characterized by slow degradation over time, which becomes significant after 24 h. At the same time, the poor stability of some esters (e.g., **2**, **4**, **6**, **14**, and **16**) is mainly due to hydrolysis of the ester group under the action of esterases present in human serum. Indeed, the capacity of these compounds to release H_2_S was almost equal to that of the corresponding acids. On the other hand, the aromatic esters (**10**, **12**, and **18**) were more stable, and consequently, their kinetics of hydrolysis was different from that of corresponding acids (**9**, **11**, and **17**). It seems that the absence of an ionizable carboxyl group facilitates the release of H_2_S from H_2_S donors. Generally speaking, the H_2_S released from all the studied molecules turned out to be gradual and prolonged in time. From these experiments, it is possible to confirm that the series of synthesized compounds cover a wide range in terms of the quantity of H_2_S released.

### 3.2. Functional Studies

#### 3.2.1. Vasodilating Activity

The vasodilator action of H_2_S-releasing compounds was evaluated on endothelium-denuded rat aorta strips precontracted with KCl. The results reported in Table 1 show the heterogeneous behavior of the tested compounds. For the most active compounds, vasodilation was found to be concentration dependent, with EC_50_ values ranging from 20 to 78 μM. In experiments performed in the presence of ATP-modulated K^+^-channels (K^+^_ATP_ channels) blocker—glibenclamide the concentration–response curves obtained for compounds **4**, **6**, and **9** were significantly rightward shifted. These results confirmed that the relaxation induced by these compounds was mediated by the activation of the K^+^_ATP_ channels. On the contrary, for compounds **8** and **12**, the EC_50_ values obtained in the presence and absence of glibenclamide were almost the same, suggesting that these compounds exerted vasorelaxation with a different mechanism(s). An example of the vasodilation profile of the H_2_S donors tested is reported in Figure 2. For the less potent compounds, the percent of relaxation at the maximal testable concentration is reported in Table 1; due to solubility, the maximal concentration tested varied from 10 to 300 μM.

#### 3.2.2. Effect of H_2_S Donor on Acetylcholine-Induced Vasodilation in Aorta Incubated with Pyrogallol

Superoxide anion (O_2_^−^) generated by pyrogallol significantly reduced maximum Ach-induced relaxation in aortic rings from 80 ± 3% to 15 ± 3% (*p* = 0.0077, *t*-test). Treatment with catechin (10 mM), a known O_2_^−^ scavenger [38], restored Ach-induced relaxation to control levels (77 ± 3%; Figure 3).

Treatment with compound **6** (100 μM) without exposure to pyrogallol had no effect on maximum relaxation (83 ± 6%; Figure 3), confirming that the amount of H_2_S released did not impair endothelium function. Pre-incubation with compound **6** attenuated the relaxation dysfunction induced by pyrogallol exposure, with a significant increase in Ach-induced maximal relaxation to 48 ± 11% compared to 15 ± 3% in the pyrogallol group (*p* = 0.0085, *t*-test; Figure 3). On the contrary, compound **8** (100 μM) was not able to reduce pyrogallol-induced endothelium impairment (Appendix A).

## 4. Discussion

Experimental evidence has increasingly shown that alteration in H_2_S production is connected to cardiac pathologies. H_2_S has been hypothesized to have a protective role against the onset and development of atherosclerosis. While failures in H_2_S signaling, including the enzymes that synthesize it, can lead to cardiovascular diseases (CVD) and associated complications, H_2_S-based interventions have proved to be helpful in avoiding adult-onset CVD in animal studies by reversing disease-programming processes [6]. Indeed, the cardiovascular system can be programmed by a series of early-life offenses, driving to CVD in adulthood. Various models of developmental programming have been studied, including the genetic and maternal hypertension model, the suramin-induced preeclampsia model, maternal nicotine exposure, and high-fat diet: in these models, the reprogramming effects of H_2_S-based interventions have been reported in rats ranging from 12 weeks to 8 months of age, which approximately corresponds to human ages from young to middle adulthood [39].

However, the fine balance of H_2_S amount coming from endogenous production or exogenous H_2_S-releasing agents is very important. Similar to other gaseous transmitters, H_2_S has a double-face behavior with beneficial effects at low concentrations but potentially damaging effects at higher doses. The balance between endogenous H_2_S synthesis and exogenous H_2_S-releasing compounds that can have an effect on the fine H_2_S balance is important and requires a careful examination of the complex relationship between H_2_S and CVD.

Although many compounds have been synthetized and studied through in vitro and in vivo experiments, displaying H_2_S-release and positive effects in the treatment of cardiovascular diseases, to date, it is still not possible to identify an optimal compound, due to some drawbacks for each of them. In fact, the major limits of endogenous H_2_S-donor molecules are their incapacity to imitate endogenous H_2_S generation together with reactive byproducts formation, the nature and biological activity of which are frequently unknown [33]. A great number of H_2_S releasing molecules have been developed and evaluated, including N-(benzoylthio)benzamides [40], acyl perthiols [41,42], arylthioamides [43], 1,2,4-thiadiazolidin-3,5-diones [44], iminothioethers [45], mercaptopyruvate [46], dithioates [47], isothiocyanate [48], and thiocarbamates [49].

In this work, we decided to investigate various classes of H_2_S donors with different activation mechanisms as well as different kinetics and amounts of released H_2_S. All of the designed molecules were furnished with a carboxyl group that allows the desired H_2_S donors to be conjugated to other pharmacophores in the future. The corresponding alkyl esters were also studied to exclude the possibility of ionization and increase the lipophilicity of the studied molecules. In particular, S-allyl sulfide, S-allyl disulfide, benzylthioamide, N-(aryloylsulfanyl)acetamide, 3H-1,2-dithiole-3-thione and 1,3-dithiole-2-thione substructures were taken in account (Figure 1).

The H_2_S donors showed a very varied and complex vasodilating profile. Among the series, compounds **4**, **6**, **8**, and **12** were found to be the most potent vasodilators (Table 1). There is no direct correlation between the quantity and velocity of H_2_S release and vasodilatation. For example, a good H_2_S releaser bearing the thioamide substructure **18** was a less potent vasodilator than 3H-1,2-dithiole-3-thione derivative **9** with rather poor H_2_S production. In the acid/ester pair, the better H_2_S production of the ester derivatives results in a better vasodilatation profile, with the only exception of **9**/**10**. It seems that for effective vasorelaxation, H_2_S release should occur within endothelial cells and that optimal lipophilicity is essential for intracellular penetration of the compounds. Ester derivatives have higher lipophilicity and better H_2_S-releasing properties, which result in better vasorelaxation. On the other hand, the highly lipophilic acid **9** gives rise to ester **10** with very high lipophilicity, which probably hinders its intracellular accumulation resulting in a reduced vasodilating activity.

In the series of 3H-1,2-dithiole-3-thiones (**9**–**14**), the substituents present on the heteroring appear to be crucial for the potency and mechanism of vasodilation. Indeed, compound **9**, bearing two aryl substituents on 1,2-dithiole-3-thione ring, showed modest vasodilating activity, with a significant rightward shift in the concentration-response curve obtained in the presence of the K^+^_ATP_ channel blocker glibenclamide. The same behavior was observed for disulfuric compounds **4**, and **6**: the higher EC_50_ values obtained in the presence of glibenclamide (Table 1, Figure 2a) confirmed that the vasorelaxation induced by these compounds was due to the opening by H_2_S of K^+^_ATP_ channels by sulfhydration of the Kir6.1 subunit, which is thought to be the main mechanism underlying the vasorelaxant effects of H_2_S [50].

Removing the aryl from the 5-position of the 3H-1,2-dithiole-3-thione ring yields the best vasodilator in the series, compound **12**. This compound probably combines an optimal lipophilic profile with good H_2_S-releasing capacity, but its vasodilation has a different mechanism. Indeed, the EC_50_ values obtained in the presence and absence of glibenclamide were almost the same, suggesting that the observed vasorelaxation was not caused by the activation of K^+^_ATP_ channels. The same behavior was observed for compound **8**, bearing an N-mercapto substructure (Table 1; Figure 2b). Both of these derivatives released more H_2_S than the other compounds in the library, but the vasodilating effects were not induced by the main vasorelaxant mechanism of H_2_S. It is known that the K^+^_ATP_ channels’ activation is not the only mechanism by which H_2_S exerts vasodilation; Bucci et al. [51] demonstrated that H_2_S could act as an endogenous nonselective phosphodiesterase inhibitor that raises cyclic nucleotide levels in tissues, causing vasodilation. Additional ion channel-independent mechanisms include, among others, decreased ATP levels through metabolic inhibition [52] and intracellular pH decrease exerted by activation of the chloride/bicarbonate exchanger [53], albeit H_2_S-induced stimulation of the type 2 anion exchanger of vascular smooth muscle cells, resulting in bicarbonate influx and superoxide anion efflux, has been connected to nitric oxide inactivation and vasoconstriction [54]. Moreover, it was demonstrated that not only K^+^_ATP_ channels but also K_v_7 channels (particularly the K_v_7.4 subtype) are relevant pharmacological targets for H_2_S, and that direct activation of this family of ion channels mediates an important part of the vasodilatory effects of this gaseous mediator [55]. The relative contribution of K^+^ channels versus alternative pathways in the vasorelaxation due to H_2_S is expected to vary depending on the vascular bed, the species studied, and the amount of H_2_S present in the microenvironment [56], thus justifying the K^+^_ATP_-insensitive dilatory response of compounds **8** and **12**.

Shifting the aryl substituent from the 4 to 5 position of the heteroring leads to molecules that are less active (**13**, **14**) as both H_2_S releasers and vasodilators.

Endothelial dysfunction is associated with the onset of different CVDs, such as hypertension, myocardial infarction, cardiovascular complications of diabetes, and atherosclerosis. H_2_S could have a beneficial effect by triggering the angiogenesis of endothelial cells and relaxing vascular smooth muscle cells, thereby decreasing blood pressure [57]. Indeed, a low level of H_2_S endogenous production is implicated in the pathogenesis of endothelial dysfunction, while the administration of exogenous H_2_S, using H_2_S donors, can help to recover endothelial function and slow down the progression of CVD [27]. Oxidative stress is one of the most important mechanisms involved in endothelial dysfunction and associated cardiovascular pathogenesis. Therefore, reducing endothelial cells’ injury caused by oxidative stress could be a key to the treatment and/or prevention of CVD. Wen et al. reported the capacity of H_2_S to protect endothelial cells against oxidative stress due to its direct antioxidant activity, as well as its role in maintaining mitochondrial integrity [32]. There are still some uncertainties about the exact mechanisms of the antioxidant action of H_2_S, although different hypotheses have been proposed. The family of silent information regulator 2 (SIR2) is functionally significant in endothelial cells under oxidative stress, and some works point out that H_2_S is able to tune the activity of the sirtuin family, as the upregulation of sirtuin1 (SIRT1) in human PC12 cells and human umbilical vein endothelial cells (HUVECs) and the rise of SIRT3 and sirtuin 6 (SIRT6), to carry out either physiological or pathophysiological effects [58]. Moreover, H_2_S (H_2_S/HS^−^/S^2−^) was helpful in biological states in which free radicals play an unfavorable role. H_2_S protects vascular smooth muscle, neurons, and different cells from oxidative stress exerted by biological or physicochemical conditions [59,60,61,62]. Indeed, the direct interaction of H_2_S with both free radicals and reactive oxygen species (ROS) have been hypothesized as a part of its antioxidant action [63,64,65].

To better understand the antioxidant activity of H_2_S in vascular tissue, we studied the ability of selected H_2_S donors to preserve NO-dependent endothelial reactivity in an in vitro model of acute oxidative stress. Probably the best-characterized mechanism by which oxidative stress can injure vascular function is the reaction of superoxide with NO [66,67], resulting in reduced bioavailability of this vasoprotective molecule. Given the ability of H_2_S to scavenge O_2_^−^, we tested whether H_2_S donors could, in turn, preserve the endogenous bioavailability of NO in isolated rat aortae treated with the O_2_^−^ generator pyrogallol to generate acute excessive levels of superoxide in vitro. Pyrogallol quickly auto-oxidizes in an oxygen-containing aqueous medium to generate O_2_^−^. It has been previously reported that this model significantly inhibits ACh-induced relaxation through a reduction in the bioavailability of endogenous NO [68]. Catechin, a well-known O_2_^−^ scavenger [38], was able to completely restore Ach-induced relaxation, albeit at high doses (10 mM) (Figure 3). For testing the ability of new H_2_S donors to protect against superoxide-mediated impairment of NO bioavailability, we selected two compounds with similar potencies but different vasorelaxant mechanisms. Compound **6** was selected as the most potent vasodilator agent via the opening of K^+^_ATP_ channels, while compound **8** was selected as one of the compounds releasing higher amounts of H_2_S but exhibiting a K^+^_ATP_-independent vasorelaxation mechanism. Both selected compounds were first tested on rat aortic rings without acute oxidative stress to verify whether the compounds could induce an impairment of NO availability. The results showed that compounds **6** and **8** (100 μM) did not reduce Ach-induced maximum relaxation (Figure 3; Appendix A), so the amount of H_2_S released by both compounds had no toxic or pro-oxidant effects.

Pretreatment with compound **6** significantly attenuated the pyrogallol-induced impairment of vasorelaxation and partially restored Ach-induced vasorelaxation (Figure 3). We suppose that the antioxidant effect of H_2_S release from **6** exerts vascular protection. The inhibitory effect of H_2_S on superoxide anion has been studied using different biochemical protocols. Polysulfides formation after several minutes incubation in buffered stock solutions of Na_2_S or NaHS at pH 7.4 have been demonstrated; Misak et al. showed that the formation of polysulfides and other H_2_S oxidation products is time-dependent when freshly prepared Na_2_S in H_2_O is diluted in Tris-HCl buffer, pH 7.4 [68]. Thus, we may assume that under our experimental conditions, polysulfides and other sulfide derivatives might be formed during incubation of the H_2_S donor solution in organ baths, and the observed protective effects of H_2_S could be affected by the formation of polysulfides or other H_2_S derivatives.

Although **8** is a more effective H_2_S donor, this compound tested under the same experimental conditions did not protect the endothelium from O_2_^−^-induced damage. The lack of protection cannot be due to a too high amount of H_2_S being released into the organ bath since incubation of **8** alone had no negative effects on Ach-induced vasorelaxation (Appendix A). This behavior may be related to the H_2_S release mechanism for compound **8**. Indeed, the release of 1 mole of H_2_S from **8** requires an excess (3 moles) of sulfhydryl groups, mainly glutathione (GSH), with GSSG formation [40,69]. GSH is the most abundant natural cellular antioxidant and plays a key role in maintaining the cellular redox balance. The antioxidant effect of GSH is due to the formation of its oxidized form: glutathione disulfide (GSSG). The latter is reduced to GSH by glutathione reductase to maintain a proper cellular GSH/GSSG ratio [70]. The H_2_S release process from compound **8** can significantly impair the GSH/GSSG ratio in endothelial cells. At the same time, in the experimental pyrogallol model, the incubation time of the H_2_S donor before the induction of endothelial damage was too short to allow for the recovery of glutathione lost, and the amount of H_2_S released was not able to compensate for the reduction in GSH/GSSG ratio and the consequent alteration of the cellular redox state. On the other side, the S-allyl disulfide substructure of compound **6** has been demonstrated to up-regulate the glutathione level by enhancing the expression of the regulatory subunit of glutamate-cysteine ligase (GCLM) and to increase the expression of phase II detoxifying enzymes, improving protection from oxidative stress [71,72].

This work provided a small library of H_2_S donor molecules with an extensively varied chemical scaffold showing different H_2_S-releasing rates and amounts, and vasodilating profiles. In particular, compound **6** was shown to vasodilate rat aorta strips through K^+^_ATP_ channels activation and, more interestingly, was also able to protect vascular endothelium from oxidative damage. Therefore, this work could allow rationalizing of the development of H_2_S-donor molecules that are potentially useful for the treatment of cardiovascular diseases.

## Figures and Tables

**Figure 1 antioxidants-12-00344-f001:**
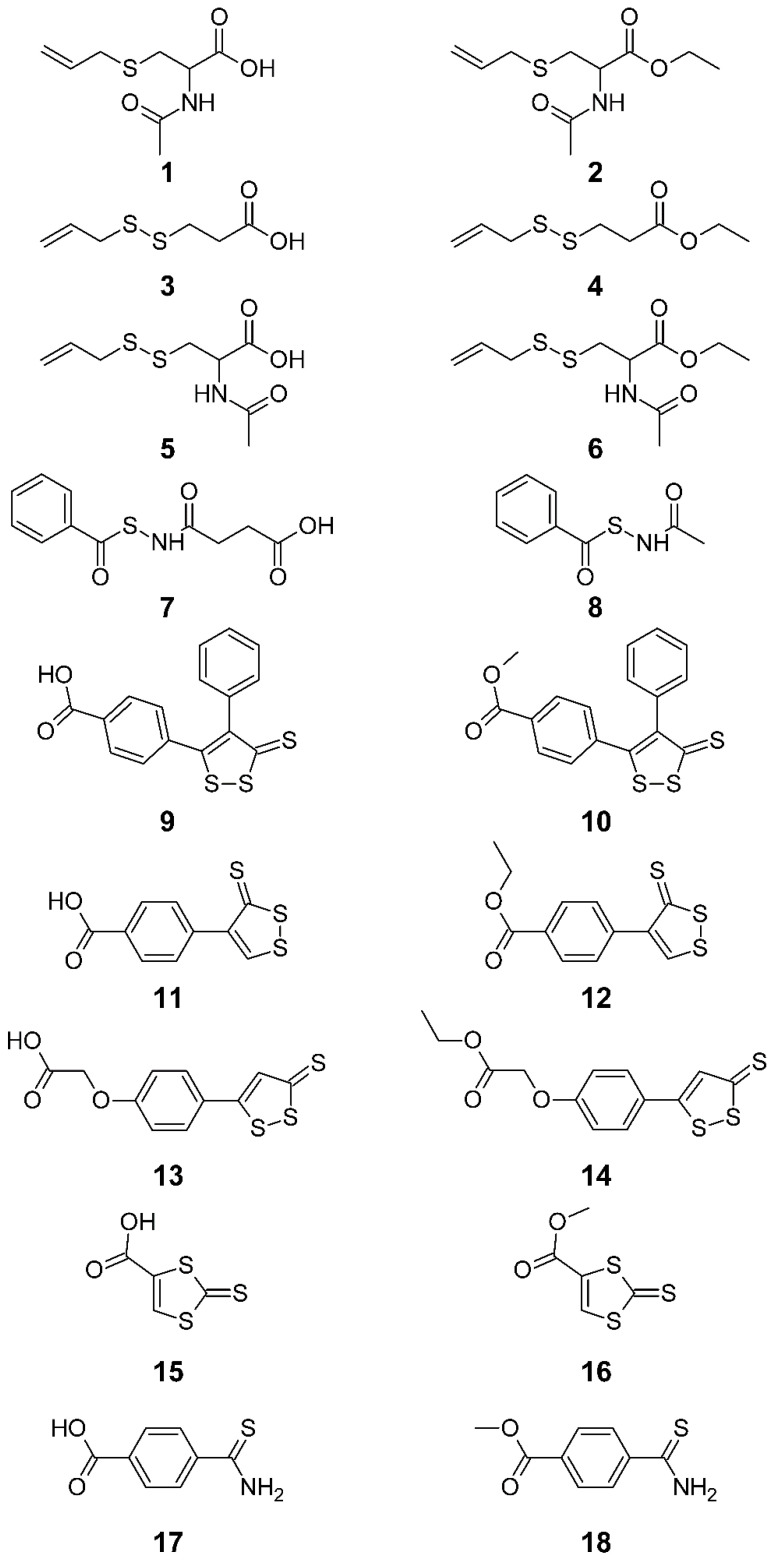
Structures of H_2_S donor molecules.

**Figure 2 antioxidants-12-00344-f002:**
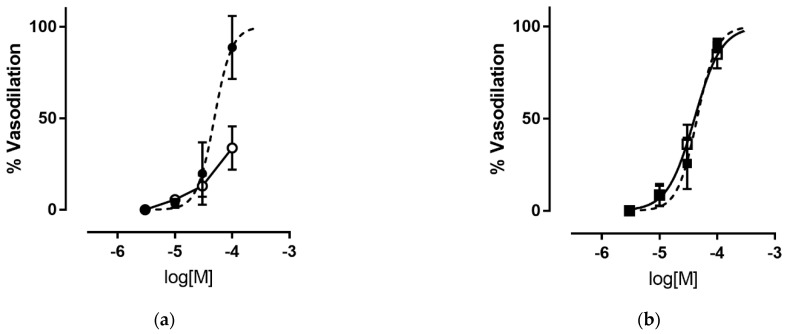
Vasodilating effects on rat aorta strips precontracted with 25 mM KCl: (**a**) **6** (dotted line, ●), **6** and 10 μM glibenclamide (straight line, ○); (**b**) **8** (dotted line, ■), **8** and 10 μM glibenclamide (straight line, □).

**Figure 3 antioxidants-12-00344-f003:**
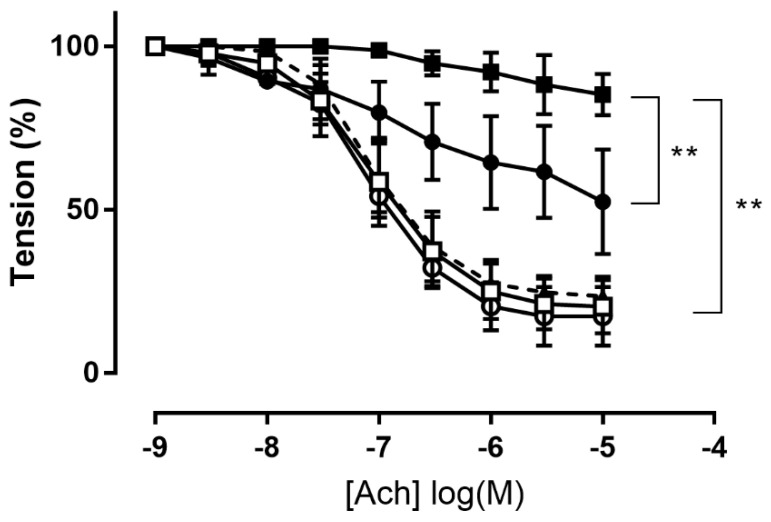
Effect of **6** (100 μM) on acetylcholine-induced vasodilation in endothelium-intact rat aortic rings. Control: rings incubated with vehicle only (DMSO) (straight line, □); rings incubated with **6** (straight line, ○); rings incubated with 500 μM pyrogallol (straight line, ■); rings incubated with **6** and 500 μM pyrogallol (straight line, ●); rings incubated with catechin (10 mM) and 500 μM pyrogallol (dotted line, ∆). Statistical analyses were made using Student’s *t*-test for unpaired data. ** *p* < 0.01 vs. the pyrogallol group.

**Table 1 antioxidants-12-00344-t001:** Stability, H_2_S release, and vasodilating activity of H_2_S donors.

Compd ^1^	Stability and H_2_S Release In Human Serum				Vasodilating Activity	
	% Compd (at 1, 4, 24 h) ± SE or Half-Life		% H_2_S mol/mol (at 1, 4, 24 h) Hours ± SE		EC_50_ (μM) ± SEM [% Vasodilation]	+ Glib. 10 μM EC_50_ (μM) ± SEM [% Vasodilation]
**1**	88 ± 1	1 h	0.4 ± 0.1	1 h	NA	NA
	35 ± 1	4 h	1.5 ± 0.1	4 h		
	8.0 ± 0.5	24 h	3.9 ± 0.9	24 h		
**2**	t_1/2_ < 30 min		0.6 ± 0.3	1 h	NA	NA
			1.8 ± 0.3	4 h		
			5 ± 1	24 h		
**3**	80.0 ± 1.0	1 h	1.5 ± 0.2	1 h	NA	NA
	62 ± 2	4 h	8.2 ± 0.3	4 h		
	44 ± 1	24 h	23 ± 2	24 h		
**4**	t_1/2_ < 30 min		5.8 ± 0.2	1 h	64 ± 6	[44 ± 3] ^2^
			10 ± 1	4 h		
			24.2 ± 0.9	24 h		
**5**	64 ± 1	1 h	2.7 ± 0.2	1 h	[16 ± 4] ^3^	
	39 ± 1	4 h	11 ± 2	4 h		
	16 ± 1	24 h	21 ± 1	24 h		
**6**	t_1/2_ < 30 min		5.8 ± 0.2	1 h	47 ± 8	[34 ± 7] ^2^
			10 ± 1	4 h		
			24.2 ± 0.8	24 h		
**7**	19.8 ± 0.7	1 h	0.3 ± 0.2	1 h	[12 ± 1] ^3^	
	0	4 h	1.4 ± 0.2	4 h		
	0	24 h	5 ± 1	24 h		
**8**	t_1/2_ < 30 min		16 ± 1	1 h	40 ± 3	40 ± 4
			25.8 ± 0.4	4 h		
			47 ± 1	24 h		
**9**	94.2 ± 0.2	1 h	0.4 ± 0.1	1 h	78 ± 7	262 ± 62
	90.0 ± 0.8	4 h	1.7 ± 0.6	4 h		
	83 ± 1	24 h	4.4 ± 0.9	24 h		
**10**	t_1/2_ = 2.4 h		1.8 ± 0.3	1 h	[29 ± 3] ^4^	
			7.5 ± 0.6	4 h		
			12.0 ± 0.9	24 h		
**11**	96.3 ± 0.2	1 h	0.4 ± 0.4	1 h	[42 ± 3] ^2^	
	90.0 ± 0.9	4 h	4 ± 1	4 h		
	53 ± 2	24 h	33 ± 4	24 h		
**12**	t_1/2_ = 2.7 h		17 ± 8	1 h	20 ± 2	20 ± 3
			63 ± 30	4 h		
			146 ± 34	24 h		
**13**	99.2 ± 0.1	1 h	0	1 h	[39 ± 8] ^2^	
	96.0 ± 0.8	4 h	0.7 ± 0.2	4 h		
	94.3 ± 2	24 h	2.4 ± 0.5	24 h		
**14**	t_1/2_ = 30 min		0	1 h	[37 ± 3] ^4^	
			0.5 ± 0.3	4 h		
			1.8 ± 0.6	24 h		
**15**	99.2 ± 0.1	1 h	0.2 ± 0.4	1 h	[17 ± 3] ^3^	
	95 ± 1	4 h	1.3 ± 0.2	4 h		
	94.8 ± 2	24 h	2.8 ± 0.5	24 h		
**16**	t_1/2_ = 1.5 h		0.4 ± 0.2	1 h	[25 ± 5] ^2^	
			1.7 ± 0.5	4 h		
			2.5 ± 0.9	24 h		
**17**	91.6 ± 0.3	1 h	0.2 ± 0.2	1 h	[27 ± 6] ^3^	
	85.0 ± 0.1	4 h	1.0 ± 0.3	4 h		
	64.7 ± 0.2	24 h	6 ± 1	24 h		
**18**	t_1/2_ = 3.7 h		1.4 ± 0.4	1 h	[33 ± 6] ^4^	
			11.0 ± 0.3	4 h		
			44 ± 2	24 h		

^1^ Compound. ^2^ Percent vasodilation at the maximum concentration testable (100 μM). ^3^ Percent vasodilation at the maximum concentration testable (300 μM). ^4^ Percent vasodilation at the maximum concentration testable (10 μM).

## Data Availability

Data supporting the findings of this study are available from the corresponding authors upon reasonable request.

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
