# Peer review of "Comparative Study of Different H2S Donors as Vasodilators and Attenuators of Superoxide-Induced Endothelial Damage"

_antioxidants, 2023, doi:10.3390/antiox12020344_

Round 1

Reviewer 1 Report

In the original article entitled “Comparative study of different H2S donors as vasodilators and attenuators of superoxide-induced endothelial damage” by dr Marini et al., the Authors synthesized a series of novel H2S donors and tested their vasodilatory potential using functional studies in animal model. In my opinion, the article reports interesting results however some points listed below should be addressed.

Major comments

1.      „2.3. Functional experiments” The crucial information on laboratory animal species as well as strain are missing.

2.      „2.3.3 Data analysis” Did the Authors verify a normality of data distribution? It is a prerequisite to perform parametric tests such as Student t test.

Minor comments:

1.      Page 5, “Animals were handle” -> Animals were handled

2.      Page 5 “6.23.2018” Please correct the date

Author Response

Authors are grateful for the reviewer for the valuable comments. We performed requested corrections of manuscript:
1.    2.3. “Functional experiments” The crucial information on laboratory animal species as well as strain are missing.
We have added missing information.
2.    “2.3.3 Data analysis” Did the Authors verify a normality of data distribution? It is a prerequisite to perform parametric tests such as Student t test.
- Yes, we have verified gaussian distribution of data using the D’Agostino-Pearson normality test.
Minor comments:
1.    Page 5, “Animals were handle” -> Animals were handled
correction done
2.     Page 5 “6.23.2018” Please correct the date
correction done

Reviewer 2 Report

I believe that the manuscript written by Elisabetta Marini et al., titled “Comparative study of different H2S donors as vasodilators and attenuators of superoxide-induced endothelial damage” is the result of a carefully designed and executed experimental study where the researchers have studied the H2S-releasing capacity of a high number of H2S donor molecules under various mechanisms. Also, the endothelium-protective capacities of select compounds have been studied under induced oxidative stress conditions in an in vitro model, which is great. In general, I believe this is a high-quality work whose scientific value should be appreciated. I would like to thank the authors for undertaking this research and generating this manuscript, and I have only a few suggestions/ comments to make to help improve the quality of the manuscript. My comments are as it follows:

1. On page 4, sub-section 2.2.3 RP-HPLC analysis of stability assays, line 10 of the paragraph, in the sentence: “... 80 to 35% acetonitrile between 20 and 25 min”, is last 35%, correct? I was expecting it to be more than 80% given the increasing concentrations of acetonitrile as the mobile phase.

2. In Table 1, please write the full name for the abbreviation “Compd” in the table footer.

3. In Table 1, the last two columns of the table, some numbers have the sign % but others don’t. This must be done consistently (either use all of them with the % sign or without the sign). Having said that, since it is mentioned in the footer that the numbers of percentages, I don’t think using the % sign is necessary in the table, but again it is up to that authors to decide (as long as they do it consistently). Thank you.

Author Response

Authors are grateful for the reviewer for the valuable comments. We performed requested corrections of manuscript: 

1.    On page 4, sub-section 2.2.3 RP-HPLC analysis of stability assays, line 10 of the paragraph, in the sentence: “... 80 to 35% acetonitrile between 20 and 25 min”, is last 35%, correct? I was expecting it to be more than 80% given the increasing concentrations of acetonitrile as the mobile phase.
The decrease of the acetonitrile concentration is needed to return the chromatographic column into original (pre-eluition) condition. Indeed, this part of gradient program has nothing to do with eluition, so we removed it from the text.
2.    In Table 1, please write the full name for the abbreviation “Compd” in the table footer.
- correction done
3.    In Table 1, the last two columns of the table, some numbers have the sign % but others don’t. This must be done consistently (either use all of them with the % sign or without the sign). Having said that, since it is mentioned in the footer that the numbers of percentages, I don’t think using the % sign is necessary in the table, but again it is up to that authors to decide (as long as they do it consistently). 
We have changed the last column of the table in order to make it more clear.